# Effects of a Clinical Simulation Course about Basic Life Support on Undergraduate Nursing Students’ Learning

**DOI:** 10.3390/ijerph18041409

**Published:** 2021-02-03

**Authors:** María del Mar Requena-Mullor, Raquel Alarcón-Rodríguez, María Isabel Ventura-Miranda, Jessica García-González

**Affiliations:** 1Department of Nursing, Physiotherapy and Medicine, Faculty of Health Sciences, University of Almería, 04120 Almería, Spain; mrm047@ual.es (M.d.M.R.-M.); ralarcon@ual.es (R.A.-R.); 2Department of Nursing, Faculty of Health and Social Sciences, Campus de Lorca, University of Murcia, 30800 Murcia, Spain; jessyka_gg@um.es

**Keywords:** cardiopulmonary resuscitation, basic life support, nursing education, simulation, learning, skills

## Abstract

Training in basic life support (BLS) using clinical simulation improves compression rates and the development of cardiopulmonary resuscitation (CPR) skills. This study analyzed the learning outcomes of undergraduate nursing students taking a BLS clinical simulation course. A total of 479 nursing students participated. A pre-test and post-test were carried out to evaluate theoretical knowledge of BLS through questions about anatomical physiology, cardiac arrest, the chain of survival, and CPR. A checklist was used in the simulation to evaluate practical skills of basic CPR. The learning outcomes showed statistically significant differences in the total score of the pre-test and after completing the BLS clinical simulation course (pre-test: 12.61 (2.30), post-test: 15.60 (2.06), *p* < 0.001). A significant increase in the mean scores was observed after completing the course in each of the four parts of the assessment protocol (*p* < 0.001). The increase in scores in the cardiac arrest and CPR sections were relevant (Rosenthal’s r: −0.72). The students who had prior knowledge of BLS scored higher on both the pre-test and the post-test. The BLS simulation course was an effective method of teaching and learning BLS skills.

## 1. Introduction

The real incidence of out-of-hospital cardiac arrest (OHCA) is not exactly known [1]. In Europe, the EuReCa ONE study confirms that sudden OHCA is the third leading cause of death [2]. The EuReCa TWO study describes the epidemiology of OHCA and the effects of cardiopulmonary resuscitation (CPR) before emergency medical services arrival in 28 countries in Europe [3]**.** In EuReCa TWO, overall survival in all cases in which CPR was attempted was 8%, compared with 10% in EuReCa ONE. Among patients in the Utstein comparison group, survival to hospital discharge was similar (30% in EuReCa ONE versus 31% in EuReCa TWO, respectively) [2,3]**.** This data can vary greatly between different European countries [4] and even between different regions of the same country [5]. The incidence of OHCA in the city of Wroclaw (Poland) was 102 per 100,000 inhabitants [6]. In a similar study by Daniels et al. [7] in the Udine district in Italy the incidence was higher, with 123 per 100,000 inhabitants. In the study carried out by Rosell-Ortize et al. [1], it was estimated that there were 20 cases per 100,000 habitants in Spain from 2013 to 2014. Despite significant advances in prevention, OHCA continues to be a major public health problem, as millions of people die from sudden cardiac arrest each year [8,9].

OHCA is a healthcare problem that is often associated with poor survival rates of 8–10% [3] and has not improved in 30 years [10]. Scientific evidence and the 2015 Guidelines of the European Resuscitation Council (ERC) emphasize that community response is a key factor in improving the survival rate of victims of OHCA [11,12]. For this reason, the ERC has been promoting a strategy that was endorsed by the World Health Organization: promoting cardiopulmonary resuscitation education in schoolchildren from the age of 12, with 2 h of training per year in all European countries [13].

Pasalli et al. [14] observed that nurses and doctors lacked certain knowledge in basic life support (BLS) and advanced life support (ALS) guidelines. They highlighted the fact that resuscitation training had a positive effect on theoretical knowledge of CPR. They further noted the importance of BLS and ALS training being incorporated into the nursing curriculum. Baldi et al. [8] analyzed the knowledge of cardiac arrest and CPR of medical students in their final year of study from all over Europe and confirmed that their knowledge of cardiac arrest and CPR upon graduating was limited and needed to be improved. Another study evaluated the effects of an online BLS course on nursing students’ learning and noted that it was an effective method for teaching and learning key BLS skills, in which students were able to accurately apply BLS techniques during CPR simulation [15].

Recent research conducted on university students in Malaysia found that the students had poor knowledge of hands-only CPR. These researchers highlighted the importance of increasing public awareness and understanding of CPR as an essential strategy to increase CPR success in cardiac arrest cases [8]. Various studies exist that examine the knowledge, attitude, skills, and capability of university students regarding CPR [8,11,16] and their participation of BLS in many countries [8,14,17,18]. The results suggest that the students were aware of the importance of CPR training and were willing to act regardless of their low level of knowledge [18]. Several countries have implemented CPR training at schools, including students as young as 12 years old [19]. Therefore, it is highly recommended to promote and provide CPR training to young people from an early age [20].

The 2015 American Heart Association (AHA) Guidelines include recommendations for BLS training such as simulations [15,21,22] using high-fidelity manikins, feedback devices, more frequent training, and online courses [15] as resources to teach and learn CPR skills in continuing education [23,24]. Various sources confirm that clinical simulation is an essential component of nursing education [25,26,27,28].

The aforementioned literature demonstrates that the majority of data obtained on this topic comes from research carried out in countries other than Spain [5,8,17,19]. If school-age children are familiar with CPR [13,19] then nursing students should undoubtedly be able to initiate and perform effective CPR when they begin their nursing careers [29]. For this reason, the purpose of this study was to analyze the learning outcomes of university nursing students who took a BLS clinical simulation course.

## 2. Materials and Methods

### 2.1. Study Design

A pre-post intervention study was conducted in nursing students of the Nursing Degree at the University of Murcia and University of Almería (Spain).

### 2.2. Sample and Setting

The study was carried out in two universities in the southeast of Spain, the University of Almería and the University of Murcia. The students were in their first year of the nursing degree during the 2018/2019 and 2019/2020 academic years. The total number of university nursing students was 479 (Figure 1). Nursing students who had previously participated in a clinical simulation course related to CPR were excluded from the study.

### 2.3. Variables and Measurement Instruments

The university nursing students were invited to complete a pre-test assessment protocol at the beginning of the academic year. They were asked to complete the protocol within 30 min. Sociodemographic variables were also collected, as well as other variables such as having received informative lectures on BLS to analyze its influence on learning after receiving the clinical simulation course. The anonymity of the participants was guaranteed by coding each protocol with an anonymous identifier at the beginning of the assessment test. One month after carrying out the pre-test protocols, sessions were organized with BLS clinical simulation seminars, all of which were taught by the same clinical simulation instructor. Six months after attending the BLS seminar sessions, the university students were once again invited to complete the post-test assessment protocol within 30 min. A total score was given to each participant for the BLS theoretical knowledge assessment pre-test and post- test (0 being the lowest score possible and 20 the highest).

After the course with BLS clinical simulation sessions, the university nursing students took a practical test on basic CPR. Evaluations were conducted using clinical scenarios with a simulation device that acts like a real patient. CPR clinical skills were assessed by the following: verifying the patient’s ability to respond, respiration and carotid pulse, alerting emergency services, performing CPR maneuvers (30 chest compressions/2 rescue breaths), placing their hands in the correct position, adequate depth of compressions (5 cm), correct rescue breaths, and correct use of the automated external defibrillator (AED). The time given to perform the mock practical test was 30 min. The scenarios were recorded to assess the nursing students’ performance in simulated practice settings. A score system was created by assigning one point to each correct answer, without deducting points for unanswered questions or incorrect answers. The practical simulation test was carried out by an expert in clinical simulation and was evaluated using a checklist-type list (Yes/No) (scored from 0 to 9).

### 2.4. Data Collection

The BLS learning assessment protocol was designed by expert doctors and nurses in resuscitation research and training based on the 2015 ERC guidelines of BLS and ALS. The assessment protocol for theoretical knowledge of BLS consisted of 20 questions, structured into 4 parts: questions about cardiorespiratory anatomical physiology (n = 4), questions about cardiac arrest (n = 6), questions about the chain of survival (n = 3), and questions about CPR (n = 7). The theoretical assessment protocol was completed before (pre-test protocol) and after (post-test protocol) completing the course with BLS clinical simulation sessions. Participants had 30 min to complete the assessment test.

### 2.5. Data Analysis

A database was created with the collected information. The data was analyzed using IBM SPSS 26.0 statistical software. A descriptive analysis of the continuous variables was carried out, expressed as means and standard deviations. In order to evaluate the normality of each continuous variable, the Kolmogorov–Smirnov test was performed. The related variables were analyzed using the Wilcoxon test. Rosenthal’s r was used to determine the effect size. The related categorical variables were expressed as percentages and compared using the McNemar test. The threshold of statistical significance for all tests was set at *p* < 0.05.

### 2.6. Ethical Considerations

Approval for this study was obtained from the Ethics and Research Commission of the University of Murcia (ID: 2982/2020). All the procedures were performed in accordance with the ethical standards of the Helsinki Declaration. Participation in the study was voluntary. By filling in the questionnaire, the students gave their consent to participate in the study. The questionnaire was completely anonymous and absolute confidentiality of the data provided was guaranteed.

## 3. Results

### 3.1. Characteristics of Participants

A total of 479 nursing students answered the protocol to assess their knowledge of BLS, which was divided into four parts, with questions related to anatomical physiology, cardiac arrest, the chain of survival, and basic CPR. Women accounted for 75.8% of the nursing students. The mean age was 19.84 (4.90), 100% were first-year nursing students, 98.1% had not participated in previous emergency training, 66.4% had no prior knowledge of BLS, and none had participated in a clinical simulation course before.

### 3.2. Theoretical Assessments Pre-Test and Post-Test of the Assessment Protocol of BLS Theoretical Knowledge

The learning outcomes indicated statistically significant differences in the total score of the pre-test and after completing the BLS clinical simulation course (pre-test (12.61 ± 2.30), post-test (15.68 ± 2.06), *p* < 0.001). A significant increase in the mean scores was observed after completing the simulation course in each of the four parts of the assessment protocol (*p* < 0.001). The increase in scores in the cardiac arrest and CPR sections was relevant (Rosenthal’s r: −0.72) (Table 1).

Table 2 shows the average scores of the knowledge assessment pre-test and post-test for each part of the protocol, according to gender and having prior knowledge of BLS. The analysis of the total learning score between men and women did not show statistically significant differences in the results of the pre-test (men: 12.47 (2.16), women: 12.66 (2.34), *p* = 0.45) and the post-test (men: 15.39 (1.97), women: 15.77 (2.08), *p* = 0.08). However, there was a significant increase in the total learning outcomes of both men and women between the pre-test and post-test (men: Rosenthal’s r: -0.86, women: Rosenthal’s r: −0.85). In each of the four parts of the protocol, the mean scores also increased significantly after the simulation course in both men and women (*p* < 0.001).

The participants who had prior knowledge of BLS obtained higher average scores than those who did not in both the pre-test (with training: 13.13 (2.02), without training: 12.35 (2.39), *p* < 0.001) and in the post-test (with training: 16.13 (2.05), without training: 15.45 (2.03), *p* < 0.001). In both the total learning outcomes and in each of the four parts of the protocol, a significant increase was observed in the scores obtained in the post-test compared to the pre-test (*p* < 0.001). This increase in scores was slightly higher in students who had no prior CPR training.

The percentages of correctly answered questions in each part of the protocol in the pre-test and post-test are shown in Table 3. The percentages of correct answers for each question increased in the post-test after the students completed the clinical simulation course, this increase being statistically significant for all questions (*p* < 0.001).

### 3.3. Practical Test of a Basic CPR Clinical Simulation

After the BLS simulation course, the participants took a practical test of a CPR clinical simulation, which had a mean score of 7.7 ± 1.05 (scores from 0 to 9) and was evaluated by a checklist: verified the patient’s ability to respond, verified respiration and the carotid pulse, alerted emergency services, performed CPR maneuvers (30 chest compressions/2 rescue breaths), placed their hands in the correct position, adequate depth of chest compressions (5 cm), correct rescue breaths, and correct use of the AED. The students verified the patient’s ability to respond (87.2%), verified respiration (90%), verified the carotid pulse (69.2%), alerted emergency services (98.1%), performed 30 chest compressions/2 rescue breaths (97%), placed their hands in the correct position (82.6%), performed chest compressions at least 5 cm deep (84.4%), applied two rescue breaths (95.5%), and used the AED (94.3%). No statistically significant differences were observed when comparing the mean scores obtained in the practical test between men and women (*p* > 0.05) or between students who had or did not have previous knowledge of BLS (*p* > 0.05).

## 4. Discussion

This study analyzed the learning outcomes of university nursing students who completed a BLS clinical simulation course. The students who participated in our study, similar to other nursing students in Spain, were mostly women and were approximately 20 years old [30].

The importance of this study is based on the fact that every year there are around 30,000 sudden deaths and close to 20,000 resuscitation attempts in Spain. In 2015, the ERC confirmed in its guidelines that rapid and adequate community response is vitally important to increase the survival rate of OHCA victims [11,12] Providing up-to-date information and skills training related to BLS practices is crucial for the professional development of nursing students and those with roles related to clinical practice and education [31].

Learning BLS maneuvers is considered to be very important and BLS skills can improve with practice. The AHA continues to recommend taking a BLS course to learn and practice CPR skills, including performing high-quality chest compressions. Consistent with our results, Bhanji et al. [23] confirmed that people who have received BLS training perform high-quality chest compressions and have more confidence in their abilities to do so than those who have not received training (or have not received it in the last five years).

The International Liaison Committee on Resuscitation has strongly emphasized that health professionals should receive initial BLS training before graduating. However, many healthcare students cannot demonstrate having mastered BLS upon graduation [32].

Among our results, we observed statistically significant differences in the total score of the pre-test and after completing the BLS clinical simulation course [pre-test: 12.61 (2.30), post-test: 15.60 (2.06), *p* < 0.001]. The increase in the mean scores of the nursing students after completing the BLS clinical simulation course was significant. Similar results were found by Tobase et al. [15], further emphasizing that the course was an effective method for teaching and learning key BLS skills, in which college students were able to accurately perform BLS maneuvers during the CPR simulation. In addition, a relevant increase was obtained in the scores in the cardiac arrest and CPR sections (Rosenthal’s r: −0.72) after completing the BLS simulation course. Furthermore, there are authors who stress the importance of including BLS training in nursing students’ education due to observing a lack of knowledge regarding cardiac arrest and rescue techniques [11,14].

According to the results of this study, the students who had prior knowledge of BLS obtained higher scores in both the pre-test and post-test. However, the increase in the mean score after completing the BLS clinical simulation course was significant in both men and women, similar to the findings of a previous study that measured the effect of a simulation-based educational intervention. The difference in the overall mean scores of the responses before and after the intervention were statistically significant [22].

The conclusion reached by various authors in a meta-analysis carried out on nursing educational interventions for nurses was that nursing simulation has a strong educational capacity [33]. Clinical simulation of cases with real patients is being developed at all universities, including the two that participated in this study. As in our case, there are universities that have several widely equipped simulation classrooms where students can learn various medical disciplines, including nursing. Its success lies in the fact that this form of learning gives students another perspective on medical practice and situations that they will have to face in the near future [22,26].

The reason clinical simulation was chosen in our study to improve the BLS knowledge of nursing students is because there are authors who pointed out that among the advantages of this teaching method is that it allows students to actively participate in the learning process in order to increase their nursing performance ability based on clinical cases [34,35]. In addition, previous research was found that confirms that clinical simulation practice contributes to improving the ability to solve problems, clinical performance, and nursing students’ knowledge, as well as a certain degree of learning satisfaction [35,36,37].

Additionally, the 2015 AHA guidelines state that it is wise to employ high-fidelity simulators and feedback devices to improve the quality of CPR [35,38]. For this reason, this type of BLS clinical simulation course was used in our investigation because there is scientific evidence that confirms that students who use BLS simulations, high-fidelity manikins, and other devices that provide corrective feedback during their CPR training have improved compression rates and improved CPR skills compared to those who do not use such devices [15,23]. Another study indicated that frequent practice is needed to maintain CPR skills [28]. There are several recent investigations that coincide with our results, affirming that clinical simulation is an essential component in nursing education [25,26,27,28].

### Limitations

The generalizability of the results is limited, as nursing students from only two universities in the southeast of Spain were studied. In addition, the amount of prior knowledge of CPR among nursing students differed in this study. Finally, our conclusion may not represent all nursing students from Spain, as our analysis was from a limited number of universities with nursing programs. It would be advisable to carry out studies with a control group and triangulation with qualitative studies that delve into aspects that go beyond the dimensions included in this study.

## 5. Conclusions

The BLS clinical simulation course was an effective method of teaching and learning BLS skills, in which university nursing students were able to rigorously perform BLS procedures during simulated clinical CPR practice. This type of training can improve the learning outcomes of nursing students and promote continuing education for health professionals. Repeating BLS training throughout nursing education is recommended in order to increase the effectiveness of BLS practices during training in clinical and real-life situations.

## Figures and Tables

**Figure 1 ijerph-18-01409-f001:**
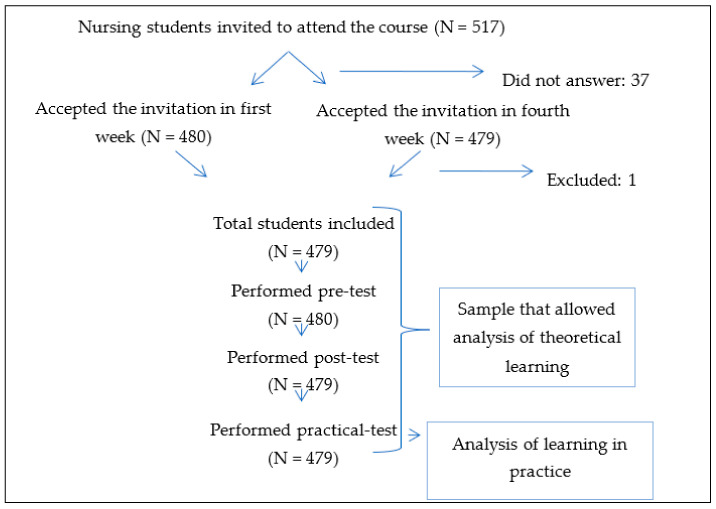
Flow diagram of the participants.

**Table 1 ijerph-18-01409-t001:** Description of pre- and post-test scores by parts of the protocol.

Parts of the Protocol	Pre-Test *	Post-Test *	Within-Group Change Scores	*p*-Value ^†^	r de Rosenthal
Learning total	12.61 ± 2.30	15.68 ± 2.06	−3.06 (−3.2, −2.89)	0.001	−0.85
Anatomy-physiology	2.54 ± 0.84	3.12 ± 0.80	−0.58 (−0.64, −0.51)	0.001	−0.63
Cardiac arrest	4.13 ± 1.13	5.06 ± 0.96	−0.93 (−1.01, −0.85)	0.001	−0.72
Chain of survival	1.39 ± 0.89	1.87 ± 0.88	−0.47 (−0.54, −0.41)	0.001	−0.53
CPR	4.55 ± 1.38	5.62 ± 1.09	−1.07(−1.16, −0.97)	0.001	−0.72

* Values are expressed as mean ± SD for pre-test and post-test means and as mean (95% CI) for within-group change scores. ^†^
*p*-value obtained by Wilcoxon test. BLS = basic life support; CPR = cardiopulmonary resuscitation.

**Table 2 ijerph-18-01409-t002:** Description of pre- and post-test scores by sex and previous knowledge in CPR.

Parts of the Protocol		Sex					Previous Knowledge in Basic CPR			
		Pre-Test *	Post-Test *	*p*-Value ^†^	r de Rosenthal		Pre-Test *	Post-Test *	*p*-Value ^†^	r de Rosenthal
Learning total	Female	12.66 ± 2.34	15.77 ± 2.07	0.001	−0.85	Yes	13.13 ± 2.02	16.13 ± 2.05	0.001	−0.85
	Male	12.47 ± 2.16	15.39 ± 1.97	0.001	−0.86	No	12.35 ± 2.39	15.45 ± 2.03	0.001	−0.86
Anatomy-Physiology	Female	2.55 ± 0.86	3.15 ± 0.81	0.001	−0.63	Yes	2.57 ± 0.91	3.14 ± 0.83	0.001	−0.61
	Male	2.53 ± 0.77	3.05 ± 0.77	0.001	−0.63	No	2.53 ± 0.80	3.12 ± 0.79	0.001	−0.63
Cardiac arrest	Female	4.14 ± 1.14	5.06 ± 0.98	0.001	−0.71	Yes	4.25 ± 1.12	5.23 ± 0.89	0.001	−0.71
	Male	4.09 ± 1.08	5.06 ± 0.91	0.001	−0.74	No	4.07 ± 1.13	4.98 ± 0.99	0.001	−0.72
Chain of survival	Female	1.40 ± 0.92	1.91 ± 0.87	0.001	−0.53	Yes	1.36 ± 0.93	1.78 ± 0.89	0.001	−0.51
	Male	1.36 ± 0.82	1.77 ± 0.89	0.001	−0.52	No	1.41 ± 0.88	1.92 ± 0.87	0.001	−0.54
CPR	Female	4.56 ± 1.38	5.65 ± 1.08	0.001	−0.73	Yes	4.95 ± 1.11	5.98 ± 0.97	0.001	−0.72
	Male	4.49 ± 1.38	5.51 ± 1.13	0.001	−0.71	No	4.34 ± 1.45	5.43 ± 1.11	0.001	−0.73

* Values are expressed as mean ± SD for pre-test and post-test means. ^†^
*p*-value obtained by Wilcoxon test. CPR = cardiopulmonary resuscitation.

**Table 3 ijerph-18-01409-t003:** Percentages of correctly answered questions in the pre-test and post-test theoretical assessments in the BLS clinical simulation course.

Questions of the Protocol	Pre-Test	Post-Test	*p*-Value ^†^
**Anatomy-Physiology**			
The place where oxygen enters the body	82.9%	92.5%	0.001
The place where oxygenation of the blood takes place	70.8%	85.4%	0.001
What happens in cardiac systole	39.5%	55.5%	0.001
Association structure-function	61%	78.9%	0.001
**Cardiac Arrest**			
When cardiac arrest occurs	70.6%	88.5%	0.001
Main mechanism that can cause cardiac arrest	73.3%	83.1%	0.001
What happens in myocardial infarction	56.6%	72.9%	0.001
Symptoms of sudden death	74.3%	91.9%	0.001
Calling in case of sudden death	70.4%	88.7%	0.001
The first thing to do when witnessing sudden death	67.8%	81.2%	0.001
**Chain of Survival**			
What to do first according to the BLS algorithm (ERC)	34.9%	51.15	0.001
Number of links in the chain of survival (ERC)	51.4%	67.8%	0.001
Second link in the chain of survival (ERC)	53.2%	68.3%	0.001
**CPR**			
Depth of chest compressions (ERC)	45.5%	67%	0.001
Frequency of chest compressions during CPR (ERC)	48%	62.2%	0.001
Objective of CPR	69.1%	83.3%	0.001
Maneuver to open the airway of victims without a cervical injury	68.9%	86.8%	0.001
Frequency of cardiac massage and mouth-to-mouth breathing	71%	84.1%	0.001
Placement of cardiac massage	71.8%	88.9%	0.001
When to stop CPR	80.4%	89.4%	0.001

Values are expressed as percentages. ^†^
*p* value obtained by McNemar test. BLS = basic life support; CPR = cardiopulmonary resuscitation; ERC = European Resuscitation Council.

## Data Availability

The data presented in this study are available on request from the corresponding author. The data are not publicly available due to privacy or ethical restrictions.

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
