# Peer review of "Effects of a Clinical Simulation Course about Basic Life Support on Undergraduate Nursing Students’ Learning"

_ijerph, 2021, doi:10.3390/ijerph18041409_

Round 1

Reviewer 1 Report

The present manuscript entitled "Effects of a Clinical Simulation Course about Basic Life Support on Undergraduate Nursing Students’ Learning "is an article in which it is observed that through clinical simulation it improves compression rates and the development of cardiopulmonary resuscitation skills. The study presents some limitations in the methodological section, since it is not mentioned whether the questionnaires used are validated or not. In addition, they should be cited and give more information.
Despite the importance of the use of clinical simulation in nursing and improvement in student learning, the article does not go further. It is limited to mentioning that through clinical simulation, compression rates and the development of cardiopulmonary resuscitation skills improve, something that is to be expected and that other studies already affirm.
The study does not provide relevant knowledge to science.

Author Response

Thank you for your valuable comments. The purpose of this study was to evaluate the learning acquired by university nursing students after taking a BLS clinical simulation course. For this reason, a validated questionnaire was not used, since knowledge was not measured even though learning was evaluated. Therefore, a BLS learning assessment protocol was designed by physicians and nurses who are experts in resuscitation research and training, based on the 2015 BLS and ALS ERC guidelines. CPR clinical skills were assessed using the practical simulation test performed by an expert in clinical simulation and assessed using a checklist (Yes / No).

Reviewer 2 Report

This article presents the impact of specific training on compression rates and the development of cardiopulmonary resuscitation (CPR) skills.

However, with such a large study group, the use of ANOVA as a parametric analysis can be tempted, especially with a research plan to control the impact on gender performance and prior knowledge of CPR. By using the non-parametric Wilcoxon test, we lose information about the effect of interactions between factors (although perhaps they are not interesting for the authors).

I am also concerned about the implications of time for the results - the lack of a control group does not make it possible to estimate whether the results obtained in the study are really large or whether participation in the pre-test alone does not increase the skills of the subjects.

These are undoubtedly the limitations of the studies presented.

Author Response

  1. Although it is true that ANOVA is a more robust test and with large samples it can be verified that the differences are minimal. However, based on the classical statistical criteria, a non-parametric test has been used, as the dependent variable presents a non-normal distribution. If the dependent variable had followed a normal distribution, it would also have allowed us to carry out a multiple linear regression model to analyze the influence of the various factors on acquired learning.
  2. We greatly appreciate your contribution regarding the lack of a control group, since comparing the results obtained with those of a control group may allow us to better analyze the size of the effect obtained between the learning acquired in the pre and post-test. In lines 337-338 of the limitations, we refer to the absence of this control group. The inclusion of a control group will be considered in future simulation studies.

Reviewer 3 Report

Thank you for asking me to review this manuscript. Nurses are usually the first-responders not only in cases of in-hospital cardiac arrest but in OHCA too. Their competence in Basic Life Support is important.

I find the manuscript interesting for the community, but there are some matters that have to be addressed. Please see my comments below.

1. In my opinion, it’s worth [in line 46] to update OHCA data from the current literature from different European countries, e.g.

Gräsner J-T, Wnent J, Herlitz J, Perkins GD, Lefering R, Tjelmeland I, et al. Survival after out-of-hospital cardiac arrest in Europe - results of the EuReCa TWO study. Resuscitation. 2020;148:218–26.

Czapla, M., ZieliĹ„ska, M., Kubica-CieliĹ„ska, A. et al. Factors associated with return of spontaneous circulation after out-of-hospital cardiac arrest in Poland: a one-year retrospective study. BMC Cardiovasc Disord 20, 288 (2020)

Danielis M, Chittaro M, De Monte A, Trillò G, Durì D. A five-year retrospective study of out-of-hospital cardiac arrest in a north-east Italian urban area. Eur J Cardiovasc Nurs. 2019;18:67–74.

2. Why people with BLS course (e.g. in high school ) were not excluded from the study? You could add this information to part Materials and Methods or explain this in section limitations In Linje 333: "In addition, the amount of prior knowledge of CPR among nursing students differed in this study." Can you develop it?

3. Do you have knowledge if in period between BLS Course and post-test participant have had classes with included elements BLS? If yes You could add this information.

Author Response

1.Answer:

According to your recommendation, we have added the 3 articles that you suggest (in lines 34-41, 43-46 and 53). We have also updated and added the 3 citations in the references.

  1. Introduction

The real incidence of out-of-hospital cardiac arrest (OHCA) is not exactly known [1]. In Europe, the EuReCa-One confirms that sudden OHCA is the third leading cause of death [2] . The EuReCa TWO study describes the epidemiology of OHCA and the effects of CPR before EMS arrival in 28 countries in Europe [3]. In EuReCa TWO, overall survival in all cases in which CPR was attempted was 8%, compared with 10% in EuReCa ONE. Among patients in the Utstein comparison group, survival to hospital discharge was similar (30% in EuReCa ONE versus 31% in EuReCa TWO respectively) [2,3]. This data can vary greatly between different European countries [4] and even between different regions of the same country [5]. The incidence of OHCA in the city of Wroclaw (Poland) was 102 per 100,000 inhabitants [6]. In a similar study by Daniels et al. [7] in the Udine district in Italy the incidence was higher with 123 per 100,000 inhabitants. In the study carried out by Rosell-Ortize et al. [1], it was estimated that there were 20 cases per 100,000 habitants in Spain from 2013 to 2014. Despite important advances in prevention, OHCA continues to be a major public health problem as millions of people die from sudden cardiac arrest each year[8,9].OHCA is a healthcare problem that is often associated with poor survival rates of 8-10 % [3] and has not improved in 30 years [10].  

2.Answer:

We have not excluded it as this question was asked to know if having prior knowledge could have a negative or positive influence on the BLS clinical simulation course.

The knowledge that the participants had was small lectures received about CPR that vitiated the maneuvers, the majority had not received CPR courses.

2.2. Sample and Setting

The study was carried out in two universities in the southeast of Spain, the University of Almería and the University of Murcia. The students were in their first year of the nursing degree during the 2018/2019 and 2019/2020 academic years. The total number of university nursing students was 479 (Figure 1). Nursing students who had previously participated in a clinical simulation course related to CPR were excluded from the study.

2.3. Variables and Measurement Instruments

The university nursing students were invited to complete a pre-test assessment protocol at the beginning of the academic year. They were asked to complete the protocol within 30 minutes. Sociodemographic variables were also collected and other variables such as having received informative lectures on BLS to analyze its influence on learning after receiving the clinical simulation course.

3.Answer:

No, the participants did not receive classes with BLS elements included in the period between the BLS course and the post-test

Round 2

Reviewer 1 Report

Reject the manuscript for the comments already made.

Reviewer 3 Report

I would like to thank the authors for the work they did to improve the current manuscript.